# Spontaneous and evoked activity patterns diverge over development

**Lilach Avitan[1,2], Zac Pujic[1], Jan Mölter[1,3], Shuyu Zhu[1], Biao Sun[1], Geoffrey J Goodhill[1,3]\***

[1]Queensland Brain Institute, The University of Queensland, Brisbane, Australia; [2]Edmond and Lily Safra Center for Brain Sciences, The Hebrew University of Jerusalem, Jerusalem, Israel; [3]School of Mathematics and Physics, The University of Queensland, Brisbane, Australia

**Abstract** The immature brain is highly spontaneously active. Over development this activity must be integrated with emerging patterns of stimulus-evoked activity, but little is known about how this occurs. Here we investigated this question by recording spontaneous and evoked neural activity in the larval zebrafish tectum from 4 to 15 days post-fertilisation. Correlations within spontaneous and evoked activity epochs were comparable over development, and their neural assemblies refined in similar ways. However, both the similarity between evoked and spontaneous assemblies, and also the geometric distance between spontaneous and evoked patterns, decreased over development. At all stages of development, evoked activity was of higher dimension than spontaneous activity. Thus, spontaneous and evoked activity do not converge over development in this system, and these results do not support the hypothesis that spontaneous activity evolves to form a Bayesian prior for evoked activity.

**\*For correspondence:**
g.goodhill@uq.edu.au

**Competing interests:** The authors declare that no competing interests exist.

## Introduction

As newborn neurons mature they start to become spontaneously active (*Galli and Maffei, 1988*). Patterns of activity that are coordinated between groups of neurons then begin to form (*Blankenship and Feller, 2010*; *Luhmann et al., 2016*). Prominent examples include waves of activity in the developing mammalian retina (*Meister et al., 1991*), and neural assemblies in mammalian neocortex (*Yuste et al., 1992*) and the zebrafish tectum (*Romano et al., 2015*; *Avitan et al., 2017*). What is the reason for this youthful exuberance? One key functional role played by coordinated spontaneous activity in neural development is to assist with the formation of appropriate brain wiring (*Leighton and Lohmann, 2016*; *Ackman and Crair, 2014*). Evidence for this includes findings from multiple systems that disrupting the structure of spontaneous activity disrupts circuit development (*Kirkby et al., 2013*; *Arroyo and Feller, 2016*; *Xu et al., 2011*).

However, spontaneous activity could also play a deeper functional role: providing an internal model for the patterns of activity likely to be evoked by sensory stimuli (*Ringach, 2009*). A specific mathematical formulation is that spontaneous activity could form a Bayesian prior for stimulus-evoked activity (*Fiser et al., 2010*). If this is the case, patterns of spontaneous activity during development should gradually refine to become more similar to stimulus-evoked patterns. Evidence supporting this comes from the developing ferret cortex, where multiunit recordings of spontaneous and evoked activity in 16 neurons showed increasing similarity over development (*Berkes et al., 2011*). However, whether this principle holds at larger scale and in other species is unknown.

The larval zebrafish provides a good model system for studying these questions, due to its rapid development and ease of calcium imaging of the activity of large populations of neurons (*Sumbre and de Polavieja, 2014*). Over the period of 4–15 days post-fertilisation (dpf), spontaneous activity in the zebrafish tectum follows a specific developmental trajectory (*Avitan et al., 2016*;

*Pietri et al., 2017*). Starting at 5 dpf zebrafish start to hunt prey, with an effectiveness that increases with age (*Avitan et al., 2020*). During this same period, the neural coding of stimuli improves, as measured by the increasing accuracy with which stimulus location can be decoded from tectal activity, and increased mutual information between stimuli and responses (*Avitan et al., 2020*). However, the relationship between the changing statistical properties of spontaneous and evoked activity over development has not so far been examined. Here, we show that, despite having some similarities, spontaneous and evoked patterns diverge over development.

## Results

### Correlation structure of neural activity over development

We performed two-photon calcium imaging and recorded tectal spontaneous and stimulus driven activity in fish aged 4, 5, 8–9, and 13–15 dpf (n = 11, 15, 10, 6 respectively; for convenience, we hereafter refer to the latter two groups as just 9 dpf and 15 dpf respectively) (*Figure 1A*). A similar average number of neurons was recorded at each age (*Figure 1—figure supplement 1*). We recorded 30 min of spontaneous activity in the dark (we refer to this period as SA) followed by 61.6 min of evoked activity. Visual stimulation consisted of 20 repetitions of nine 6° spots (a size likely to be interpreted as prey [*Bianco et al., 2011*] and a frequent stimulus of behavioural relevance for the larvae) at different positions covering 120° of the larvae's visual field. Spots were presented for 1 s each with a 20 s gap between spots, for a total of 180 spot presentations per fish. We refer to the whole of the visual stimulation period as TEA ('total evoked activity'). For some later analyses, we subdivide this into the activity occurring within the first 5 s of stimulus presentation, referred to as EA ('evoked activity'), and the activity occurring between 6 and 20 s post stimulus presentation, referred to as SE ('spontaneous within evoked activity') (*Figure 1B*).

Qualitatively similar activity was present in the absence of visual stimuli (i.e. in the dark), locked in response to visual stimuli, and between stimuli presentations (*Figure 1C*). To quantify these relationships, we first examined similarity in pairwise correlation profiles between spontaneous and evoked activity over development. Correlations decreased with distance for both SA and TEA, with higher short-range correlations for TEA compared to SA (*Figure 1D*). At the population level, neuron-neuron correlation matrices for each fish showed qualitative similarities in their structure (*Figure 1E*). We then calculated the correlation coefficient between each pair of correlation matrices (TEA and SA) and found no difference correlation between matrices over development (*Figure 1F*). Thus, in terms of correlational structure spontaneous and evoked activity did not become more similar over development.

### Spontaneous and evoked assemblies undergo similar developmental changes

Neural activity is often structured into assemblies, that is, recurring groups of coactive neurons. Using a graph theory-based technique for assembly detection (*Avitan et al., 2017*), recently shown to have state-of-the-art performance in detecting neural assemblies in calcium imaging data (*Mölter et al., 2018*), we identified reoccurring assemblies separately during SA and TEA (*Figure 2A*). SA and TEA assemblies did not differ in the number of neurons per assembly (*Figure 2B*). However, the spatial coverage of SA assemblies was greater than TEA assemblies (*Figure 2—figure supplement 1A–B*).

Assemblies were spatially clustered along the tectal anterior-posterior (AP) axis. Computing the centre of mass (CoM) for each assembly and projecting it onto this axis revealed that assemblies exhibited a posterior shift from 5 to 15 dpf in both TEA (*Figure 2C*) and SA (*Figure 2D*). Neurons within each assembly tended to be tuned for similar regions of visual space (*Figure 2F*). Assembly mean tuning (the average of the preferred stimulus of all neurons in the assembly) showed a posterior shift in the tectum, for both SA and TEA assemblies (*Figure 2G*), suggesting that both TEA and SA assemblies undergo some similar developmental processes.

Tuning variance within the assemblies showed no developmental trend (*Figure 2—figure supplement 1C*), but decreased along the AP axis (*Figure 2—figure supplement 1D*), an effect present in both SA and TEA assemblies. However, tuning variance within TEA assemblies was overall lower compared to SA assemblies (*Figure 2H*), suggesting more coherent evoked assemblies than

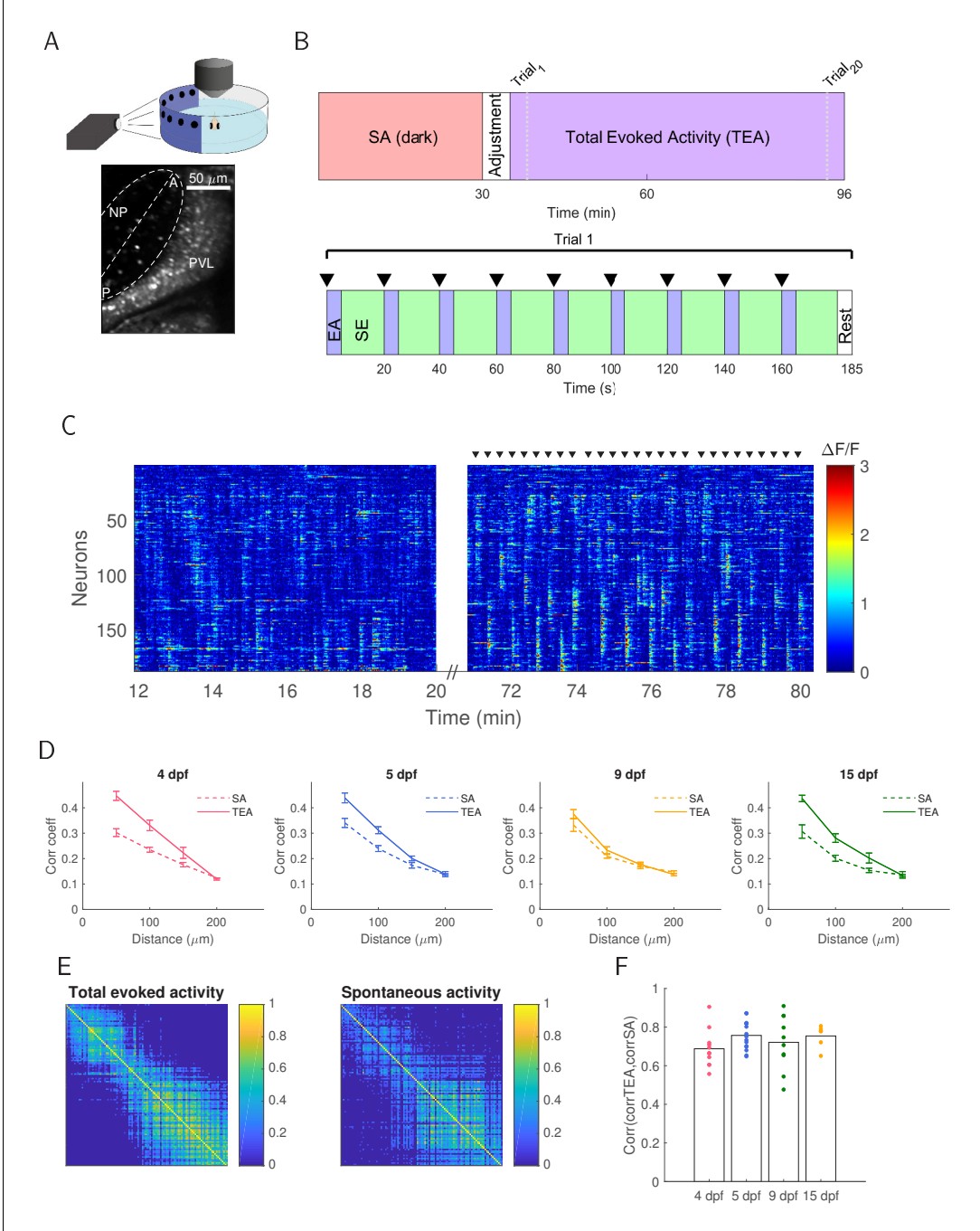

**Figure 1.** Population activity during both spontaneous and evoked activity has similar correlational structure. (**A**) Top: Larvae were embedded in agarose with one eye facing the projected image for two-photon calcium imaging. Bottom: The contralateral optic tectum (in this example 15 dpf) was imaged for 96.6 min. The neuropil (NP) contour of each fish was fitted with an ellipse (dashed line) with the major axis defining the tectal anterior-posterior axis (AP axis). Periventricular layer (PVL), NP, anterior (**A**) and posterior (**P**) ends of the tectum are indicated. (**B**) Experimental protocol. Tectal spontaneous activity (SA) in the dark was recorded after which fish were exposed to light and given 5 min to adjust. We then recorded evoked activity (TEA) in response to 20 trials of the stimulus set consisting of spots at positions 45°, 60°, 75°, 90°, 105°, 120°, 135°, 150°, 165° of the visual field (where 0° was defined as the body axis), presented in an order which maximised spatial separation within a trial. The inter-trial interval was 25 s. (**C**) Raster plot for an example 15 dpf fish showing concerted neural activity during 8 min of spontaneous activity in the dark and then during three cycles of stimulus presentation (stimulus onset is marked by black triangles). (**D**) Short-range pairwise correlation coefficients were higher for TEA compared to SA

*Figure 1 continued on next page*

*Figure 1 continued*

(4 dpf: $p = 10^{-4}$ for up to 50 µm, $p = 10^{-3}$ for 50–100 µm; 5 dpf: $p = 10^{-3}$ for up to 50 µm, $p = 10^{-3}$ for 50–100 µm; 15 dpf: $p = 10^{-3}$ for up to 50 µm, $p = 10^{-2}$ for 50–100 µm). (E) TEA and SA correlation matrices showed structural similarity (example shown is for a 15 dpf fish). Neurons were sorted by their position on the AP axis. (F) Correlation between TEA and SA correlation matrices does not change over development (one-way ANOVA, Bonferroni multiple comparison correction).

The online version of this article includes the following figure supplement(s) for figure 1:

**Figure supplement 1.** Number of active neurons recorded was similar at different ages.

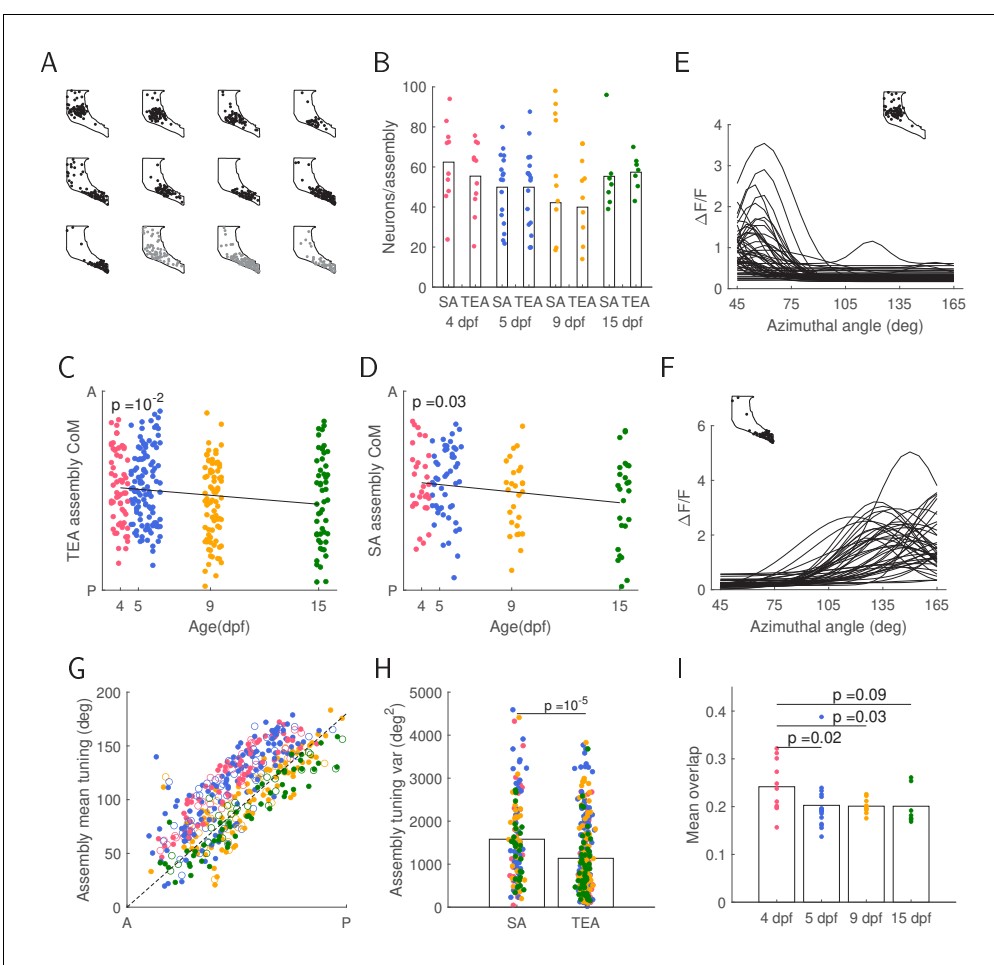

**Figure 2.** Stimulus driven and spontaneous recurring assemblies undergo similar global circuitry effects. (A) Nine total evoked activity (TEA) assemblies (black) and three spontaneous activity (SA) assemblies (gray) for the 15 dpf fish shown in *Figure 1A*. The outline shows the periventricular layer (PVL) within the field of view, with each dot representing a neuron. (B) Number of assembly neurons assigned to TEA or SA assemblies did not change over development (t-test 4 vs. 15 TEA $p = 0.5$; SA $p = 0.8$). (C-D) Centres of mass (CoMs) of assemblies within TEA or SA shifted posteriorly over development. Each column of points represents assembly CoMs for a single fish. E-F: Assembly neurons' fitted tuning curves of the most anterior (E) and posterior (F) TEA assemblies from A (inset) illustrate tuning variability within the assembly. (G) Mean assembly tuning of TEA (filled circles) and SA (empty circles) assemblies shows a posterior shift in tuning with age. Dashed line represents the perfectly linear mapping. (H) Tuning variance is higher for SA assemblies than TEA assemblies (t-test), indicating that in general tuning of spontaneous assemblies is more dispersed than stimulus driven assemblies. I: Mean overlap between TEA and SA assemblies decreases between 4 and 5 dpf and remains stable over development.

The online version of this article includes the following figure supplement(s) for figure 2:

**Figure supplement 1.** Tuning properties of neural assemblies over development.

spontaneous assemblies, driven potentially by non-identical sources of activation. The fraction of overlap in neuronal identity between TEA and SA assemblies was relatively low, ~20%, and this overlap decreased between 4 and 5 dpf and remined stable from 5 to 15 dpf (*Figure 2I*). Thus, despite some similar trends in the properties of evoked and spontaneous assemblies over development, evoked and spontaneous assemblies became progressively less similar to each other.

## Evoked and spontaneous coactivity patterns become less similar and geometrically diverge over development

We found all high-coactivity frames where the number of neurons active exceeded the number expected by chance based on a shuffle control (*Figure 3A*, see 'Materials and methods'). EA patterns recruited higher coactivity levels than either SA or SE patterns (*Figure 3B*), and these levels of coactivity within each epoch were robust over development. In addition EA patterns had higher dimensionality than either SA or SE patterns (*Figure 3C*) . Thus, in contrast to the suggestion that spontaneous activity determines the realm of evoked activity (*Luczak et al., 2009*), we found this not to be the case here. Pattern similarity for high-coactivity frames (measured by cosine distance) compared between all pairs of epochs (i.e. EA-SA, EA-SE, SA-SE) decreased over development (*Figure 3D*). This was consistent with a decrease in the overlap of neuronal identity between TEA and SA assemblies over development (*Figure 2I*).

We then projected all EA and SE patterns onto the orthogonal complement of the linear subspace spanned by SA. This measures how much of a given EA or SE pattern cannot be explained by the space spanned by SA (*Figure 3E*, see 'Materials and methods'). This unexplained fraction increased over development (*Figure 3E,F*). Similarly, the fraction of EA and SA patterns which cannot be explained by the space spanned by SE patterns also increased over development (*Figure 3G*). This result was not affected by matching the lengths of the SA and EA parts of the recording (*Figure 3—figure supplement 1*). This confirms that, over development, evoked patterns gradually increase their distance from the space spanned by spontaneous activity. Together, these results show that evoked patterns have higher dimensionality than spontaneous patterns, and become less similar and geometrically further from spontaneous patterns over development.

## Discussion

Across species, both evoked and spontaneous neural activity are characterised by time points of synchronised activity across a population of neurons (*Yuste, 2015*; *Romano et al., 2015*). Spontaneous activity in the zebrafish tectum was previously shown to be high dimensional (*Avitan et al., 2017*), similar to the mammalian cortex (*Luczak et al., 2009*). However, in the present work, evoked activity showed higher coactivity levels and higher dimensionality compared to spontaneous activity, in contrast to the suggestion that spontaneous activity outlines the realm of possible cortical sensory responses and that evoked patterns are a subset in this space (*Luczak et al., 2009*). Similarity of evoked and spontaneous patterns has been reported across species (*Berkes et al., 2011*; *Miller et al., 2014*; *Romano et al., 2015*; *Kenet et al., 2003*; *Omer et al., 2019*; *Fore et al., 2020*), with spontaneous activity being predictive of visually evoked orientation columns (*Smith et al., 2018*; *Kenet et al., 2003*) supporting the hypothesis that spontaneous activity reflects expectation of sensory experience *Luczak et al., 2009*, However, spontaneous activity in the adult cortex has been also related to the animal's ongoing behaviour (*Stringer et al., 2019*; *Carrillo-Reid et al., 2019*) and was recently shown to be orthogonal to evoked patterns (*Stringer et al., 2019*). Our results are in agreement with the latter (*Figure 3F,G*).

A developmental increase in cortical pattern similarity between evoked and spontaneous activity has been reported when evoked activity was elicited by natural stimuli, while in contrast artificial stimuli (such as gratings) resulted in a decrease in pattern similarity (*Berkes et al., 2011*). While the spot stimuli we used are quite simple, they are nevertheless highly ecologically relevant for zebrafish larvae (*Del Bene et al., 2010*; *Bianco et al., 2011*; *Preuss et al., 2014*; *Semmelhack et al., 2014*) particularly for hunting behaviour (*Bianco and Engert, 2015*). A potential explanation for the mismatch between our results and those of *Berkes et al., 2011* could be the much larger number of neurons we recorded, and that we included neurons regardless of their tuning strength.

Overall, our work shows that in the larval zebrafish tectum, despite some similarities between spontaneous and evoked activity patterns, their statistics grow more distant as development

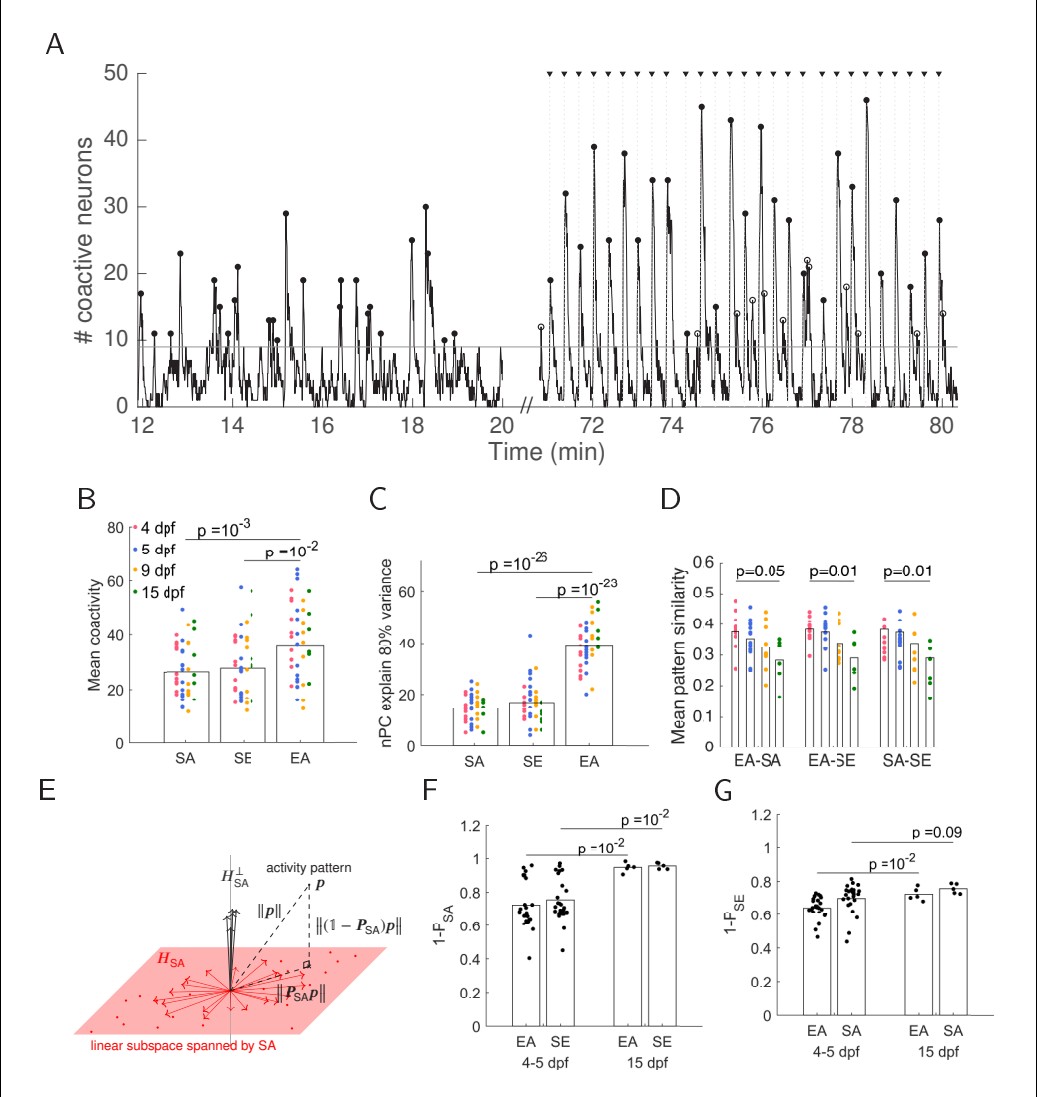

**Figure 3.** Dimensionality of evoked activity is higher than that of spontaneous activity. (A) Example showing the number of coactive neurons during 8 min of spontaneous activity in the dark and then during three cycles of stimulus presentation (stimulus onset is marked by black triangles). Significant peaks of coactivity levels ($p<0.05$, shuffled SA) during SA, EA (black dots), SE (open circles) are marked. (B) There are higher levels of mean coactivity during EA compared to SE and SA (one-way ANOVA, Bonferroni multiple comparison correction). (C) Number of principal components required to explain 80% of the variance (one-way ANOVA, Bonferroni multiple comparison correction) indicates higher dimensionality for evoked responses. In addition to the comparison shown, the dimensionality within the EA epoch increased over development; 4 (dpf) vs. 15 dpf $p = 0.05$, one-way ANOVA, Bonferroni multiple comparison correction. (D) Pattern similarity decreased between all pairs of epochs over development. Similarity was defined as the cosine similarity between the different epochs. (E) Schematic of activity patterns geometry. The patterns of a particular epoch, in this example SA patterns, were collected (red dots), and a linear subspace spanned by these patterns was calculated ($H_{SA}$, red plane; in reality this is of higher dimension than 2). Given a particular activity pattern $p$ (from a different epoch, in this example EA or SE) , $P_{SA}p$ is the projection of the pattern onto the subspace $H_{SA}$. This is the component of the pattern which can be explained by the space $H_{SA}$, and in the case where the pattern $p$ fully resides within $H_{SA}$, $P_{SA}p = p$. Conversely, $(\mathbb{1} - P_{SA})p$ is the component of the pattern which cannot be explained by the space $H_{SA}$. It is the projection of the pattern $p$ onto the orthogonal complement of the subspace spanned by SA, $H_{SA}^{\perp}$. (F) Projection of EA and SE patterns onto the orthogonal complement subspace spanned by SA patterns indicates that EA and SE patterns are less similar to SA over development. (G) Projection of EA and SA patterns onto the orthogonal complement subspace spanned by SE patterns indicates that EA and SA patterns are less similar to SE over development.

The online version of this article includes the following figure supplement(s) for figure 3:

*Figure 3 continued on next page*

*Figure 3 continued*

**Figure supplement 1.** Matching recording lengths for evoked activity and spontaneous activity does not change the conclusions.

proceeds. This result is consistent with the idea that early spontaneous activity in this system presents patterns which are similar enough to naturally evoked patterns to provide useful cues for initial wiring development, but do not attempt to track changing patterns of evoked activity over development.

## Materials and methods

### Key resources table

| Reagent type (species) or resource | Designation | Source or reference | Identifiers | Additional information |
|---|---|---|---|---|
| Genetic reagent (*Danio rerio*) | Zebrafish: Tg(elavl3:GCaMP6s) | *Vladimirov et al., 2014* | RRID:ZDB-ALT-141023-2 ZFIN Tg insertion jf5Tg | |
| Software, algorithm | MATLAB | Mathworks | https://www.mathworks.com/products/matlab.html | |
| Software, algorithm | Psychophysics toolbox | N/A | http://psychtoolbox.org/ | |
| Software, algorithm | Zen Black 2012 Service Pack 2 | Carl Zeiss Pty Ltd | http://www.zeiss.com | |

### Zebrafish

*Nacre* zebrafish (*Danio rerio*) embryos expressing *elavl3:H2B-GCaMP6s* (*Vladimirov et al., 2014*) were collected and raised according to established procedures (*Westerfield, 1995*) and kept under a 14/10 h on/off light cycle. Larvae were fed live rotifers (*Brachionus plicatilis*) daily from 5 dpf. All procedures were performed with approval from The University of Queensland Animal Ethics Committee (QBI/152/16/ARC).

### Two-photon calcium imaging

Zebrafish larvae were embedded in 2.5% low-melting-point agarose, positioned at the centre of a 35 mm diameter plastic petri dish and overlaid with E3 embryo medium. Time-lapse two-photon images were acquired at the Queensland Brain Institute's Advanced Microscopy Facility using a Zeiss LSM 710 inverted two-photon microscope. A custom-made inverter tube composed of a pair of beam-steering mirrors and two identical 60 mm focal length lenses arranged in a 4 f configuration was used to allow imaging with a 40x/1.0 NA water-dipping objective (Zeiss) in an upright configuration. Samples were excited via a Spectra-Physics Mai TaiDeepSee Ti:Sapphire laser (Spectra-Physics) at an excitation wavelength of 940 nm. Laser power at the sample plane ranged between 12 and 20 mW. Emitted light was bandpass filtered (500–550 nm) and detected with a nondescanned detector. Time-lapse images (416 $\times$ 300 pixels) were obtained at 2.2 Hz. To improve stability of the recording, chambers were left to settle prior to imaging for 3 h.

### Visual stimulation

Visual stimuli were projected onto white diffusion paper placed around the wall of the petri dish, using a pico-projector (PK320 Optoma, USA), covering a horizontal field of view of 174°. To prevent stimulus reflection, the opposite side of the dish was covered with low-reflection black paper. To avoid interference of the projected image with the signal collected by the detector, a red long-pass filter (Zeiss LP590 filter) was placed in front of the projector. Larvae were aligned with one eye facing the projected side of the dish and body axis at right angles to the projector direction. Visual stimuli were generated using custom software based on MATLAB (MathWorks) and Psychophysics Toolbox (http://psychtoolbox.org).

Larvae were imaged for spontaneous activity in the dark for 30 min after which the projector shutter was opened and larvae were given 5 min to adjust to the light conditions. We projected 20

consecutive trials of nine spots with 25 s of inter-trial interval. Each trial consisted of 6° diameter black spots at nine different positions from 45° to 165° with 15° intervals, with their order set to maximise spatial separation within a trial (45°, 120°, 60°, 135°, 75°, 150°, 90°, 165°, 105°). 0° was defined as the direction of the larvae's body axis. Spots were presented for 1 s each, followed by 20 s of blank screen.

## Automatic cell detection

Alignment of data stacks and cell detection procedure were performed as in *Avitan et al., 2017*. All fluorescence data stacks were corrected for x-y drifts using custom MATLAB software based on rigid image registration algorithm. The software automatically detected the set of pixels of each active cell and searched for pixels that showed changes in brightness across frames, resulting in an activity heat map of all the active regions across frames (*Ahrens et al., 2012*). This activity map was then segmented into regions using a watershed algorithm, with a movie-specific threshold that was similar across fish. Within each segmented region, we computed correlation coefficients of all pixels in the region with the mean of the most active pixel and its eight neighbouring pixels. Correlation coefficients showed a bimodal distribution: one peak of highly correlated pixels representing pixels of the cell within the region and a second peak of relatively low correlation coefficients representing nearby pixels within the region which were not part of the cell. Using a Gaussian mixture model, we found the threshold correlation which differentiated between pixels likely to form the active cell and neighbouring pixels that were not part of the cell. We also required that each detected active area covered at least 26 pixels (5.5 μm$^2$). The software allowed visual inspection and modification of the parameter values where needed. All pixels assigned to a given cell were averaged to give a raw fluorescence trace over time. Raw calcium signals for each cell, $F(t)$, were then converted to represent changes from baseline level, $\Delta F/F(t)$ defined as $(F(t) - F_0(t))/F_0(t)$. The time varying baseline fluorescence, $F_0(t)$, for each cell was a smoothed curve fitted to the lower 20% of the points. $F_0(t)$ was the minimum of the smoothed fluorescence trace in a 3 s window centred at $t$.

## Neuron selection

For each neuron, we calculated the mean amplitude across frames 4–7 post stimulus presentation. These amplitudes were then averaged per stimulus, providing for each neuron a curve of mean amplitude in response to all presented stimuli. Cubic spline interpolation was used to estimate the amplitude values between the presented stimuli at 5° intervals. This interpolated curve of amplitudes was fitted with a Gaussian with baseline offset. The fitted curve which provided the highest goodness of fit (adjusted $r^2$) was selected as the fit of the tuning curve. Neurons with goodness of fit greater than 0.7 were deemed to be selective neurons and included in further analysis.

## Significant correlation coefficients

We computed pairwise correlation coefficients $r$ between all pairs of neurons. To assess statistical significance, we temporally displaced each neuron's calcium trace randomly with respect to the other traces using a SHIFT algorithm as described previously (*Golshani et al., 2009*), disrupting the temporal relationship between neurons while preserving the temporal structure within each neuron. We then calculated the correlation coefficient between all pairs of shifted traces to obtain a null distribution. Pairs of neurons with a correlation coefficient greater than the 95th percentile of correlation coefficients in the null distribution were deemed statistically significant ($p<0.05$) (*Avitan et al., 2017*). We included in the correlation analysis only significant correlation coefficients.

## Neurons and assembly tuning

Assembly tuning properties (*Figure 2*) were based on tuning properties of each of the assembly member neurons. To evaluate the neurons' tuning, we calculated for each neuron the mean amplitude across frames 4–7 post stimulus presentation. These amplitudes were then averaged per stimulus, providing for each neuron a curve of mean amplitude in response to all presented stimuli. Cubic spline interpolation was used to estimate the amplitude values between the presented stimuli at 5° intervals. This interpolated curve of amplitudes was fitted with a Gaussian with baseline offset. Fit starting points used the mean as the stimulus value eliciting response peak amplitude, and the initial value for standard deviation was varied from low to high values. The fitted curve which provided the

highest goodness of fit (adjusted $r^2$) was selected as the fit of the tuning curve. Neurons with goodness of fit greater than 0.7 were deemed to be selective neurons and included in further analysis. The stimulus assigned with the peak of the fitted tuning curve was determined as the preferred stimulus of the neuron. Assembly tuning was determined as the average of the preferred stimulus over all assembly neurons.

## Selection of high coactivity patterns

The activity of each neuron was converted into a binary signal with an amplitude threshold of two standard deviations above its mean (*Avitan et al., 2017*). This provided us with a binary activity matrix where each row represents a neuron and each column represents a time bin. To establish a threshold for the significance for the number of coactive neurons, the binary activity matrix was randomly shuffled 500 times across neurons (i.e. within rows), keeping the number of 1's per cell identical, but changing their timing. The threshold corresponding to a significance level of $p<0.05$ was estimated as the number of activated neurons in a single frame that exceeded 5% of these surrogate datasets (*Miller et al., 2014*; *Avitan et al., 2017*). Every pattern which peaked in coactivity level and was above the coactivity threshold was considered in the analysis shown in *Figure 3*.

## Geometrical relation between patterns and spaces defined by patterns

We considered all high coactivity patterns for the respective types of activity (EA, SA, SE). As vectors, these patterns lie in a linear subspace $H$ ($H_{EA}, H_{SA}, H_{SE}$) within a space of all possible activity patterns. We computed an orthonormal basis for this subspace using MATLAB's orth function, which we denote as $\{u_1, \ldots, u_K\}$. With that, we have that $H = span\{u_1, \ldots, u_K\}$ and $dim H = K \leq N$. To compute the projection onto this subspace , we define the matrix $U = (u_1, \ldots u_K)$, with the vectors $u_k$ for $k = 1, \ldots K$ as column vectors. Using this matrix, $P = UU^\top$ is the projection matrix onto $H$, which for the different types of activity patterns (and corresponding subspaces ) we denote as the matrices $P_{EA}$, $P_{SA}$, and $P_{SE}$, respectively. These projections allow us to decompose any activity pattern $p$ into a part that can be 'explained' by the activity of a given type, $Pp$, and its orthogonal complement, $(\mathbb{1} - P)p$.

For example, if we consider spontaneous activity (SA) and the projection onto its subspace , $P_{SA}$, then for any given pattern of activity $p$ (such as from the pool of EA or SE patterns) $P_{SA}p$ is the part of $p$ that lies within the SA subspace , whereas $(\mathbb{1} - P_{SA})p$ lies in its complement (*Figure 3E*). In this case, we say that the larger the component $P_{SA}p$ is, the better the pattern $p$ is explained by SA, and conversely, the larger the component $(\mathbb{1} - P_{SA})p$ is, the worse the pattern $p$ is explained by SA.

We computed the fraction that could not be explained by the patterns of some type of activity. For a pattern $p$, this is the quantity $(\mathbb{1} - P)p/p$, i.e. the fraction of a pattern's total length that was orthogonal to the subspace associated with $P$. For each type of activity (EA, SA, SE), we calculated the average unexplained fraction of the high coactivity patterns corresponding to any of the two other types of activity and denote this average as '1 – $P_{EA}$', '1 – $P_{SA}$', and '1 – $P_{SE}$', respectively.

To avoid bias in generating the basis for each subspace , we selected an equal number of patterns per epoch. For consistency, this was chosen to be the minimum number of the patterns across epochs for all fish (mean of minimum number of patterns = 27 ± 12, and a minimum of 9 patterns, across 41 fish). For this number of patterns per epoch, we randomly selected them from SE and SA pools of patterns. To select EA patterns, we randomly selected an index of EA pattern and collected consecutive patterns (aiming to cover all presented stimuli with roughly similar representation). We performed these random selections of patterns 200 times, each time generating a different seed of patterns to span the spaces $H_{EA}, H_{SA}, H_{SE}$ and calculating projections of patterns onto these spaces. The distances presented in *Figure 3F and G* are means over this set.

## Sample size

Sample sizes are similar to those commonly used in the zebrafish neural imaging field.

## Acknowledgements

Imaging was performed at the Queensland Brain Institute's Advanced Microscopy Facility using a Zeiss LSM 710 2-photon microscope, generously supported by the Australian Government through the ARC LIEF grant LE130100078.

## Additional information

### Funding

| Funder | Grant reference number | Author |
|---|---|---|
| Australian Research Council | DP170102263 | Geoffrey J Goodhill |
| Australian Research Council | DP180100636 | Geoffrey J Goodhill |

The funders had no role in study design, data collection and interpretation, or the decision to submit the work for publication.

### Author contributions

Lilach Avitan, Conceptualization, Data curation, Software, Investigation, Methodology, Writing - original draft, Writing - review and editing; Zac Pujic, Investigation, Methodology, Writing - review and editing; Jan Mölter, Software, Investigation, Methodology, Writing - review and editing; Shuyu Zhu, Investigation, Methodology, Writing - original draft, Writing - review and editing; Biao Sun, Resources, Investigation, Methodology, Writing - review and editing; Geoffrey J Goodhill, Conceptualization, Supervision, Funding acquisition, Writing - original draft, Writing - review and editing

### Author ORCIDs

Lilach Avitan (iD) https://orcid.org/0000-0003-1957-8702
Jan Mölter (iD) https://orcid.org/0000-0002-5964-6207
Geoffrey J Goodhill (iD) https://orcid.org/0000-0001-9789-9355

### Ethics

Animal experimentation: All procedures were performed with approval from The University of Queensland Animal Ethics Committee (QBI/152/16/ARC).

### Decision letter and Author response

Decision letter https://doi.org/10.7554/eLife.61942.sa1
Author response https://doi.org/10.7554/eLife.61942.sa2

## Additional files

### Supplementary files

• Transparent reporting form

### Data availability

The data has been made available on figshare, under the https://doi.org/10.6084/m9.figshare.14402543.

The following previously published dataset was used:

| Author(s) | Year | Dataset title | Dataset URL | Database and Identifier |
|---|---|---|---|---|
| Avitan L, Pujic Z, Mölter J, Zhu S, Sun B, Goodhill GJ | 2021 | Spontaneous and evoked activity patterns in the larval zebrafish | https://doi.org/10.6084/m9.figshare.14402543 | figshare, 10.6084/m9.figshare.14402543 |

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
