## [Decision Letter]

**Acceptance summary:**

The relationship between evoked and spontaneous activity in neural development is an important unresolved issue and this study adds a new and interesting perspective to the existing literature. The results demonstrate that as the zebrafish develop, the spontaneous and evoked activity in the optic tectum become more dissimilar.

**Decision letter after peer review:**

Thank you for submitting your article "Spontaneous and evoked activity patterns diverge over development" for consideration by *eLife*. Your article has been reviewed by 2 peer reviewers, and the evaluation has been overseen by a Reviewing Editor and Timothy Behrens as the Senior Editor. The following individual involved in review of your submission has agreed to reveal their identity: Emre Yaksi (Reviewer #3).

The reviewers have discussed the reviews with one another and the Reviewing Editor has drafted this decision to help you prepare a revised submission.

Summary:

This manuscript investigates the relationship between spontaneous activity and evoked activity patterns in zebrafish tectum across development. The authors show that the correlation between spontaneous and evoked activity is getting weaker as animals develop. This is contrast to some of the recent findings and provide an important observation for the field. The presented data and its analysis is of high quality, and the manuscript is particularly well written an easy to read. Some of the analysis and the figures can be further improved for making the manuscript easier to read for a general audience. Reviewers also pointed out a number of methodological and statistical clarifications that need to be made as described below.

Essential revisions:

1. The part explaining the method used in figure 3 E-G should be expanded to ensure the readers can replicate the method. Specifically, it is unclear how many dimensions are necessary to describe the hyperplane H. How many basis vectors were used for the projection? Using many dimensions for the projections could bias the results towards larger angles between patterns and the hyperplane. The authors refer to [23] as the basis of their method. However, the method used in [23] is different from the method described here and puts an emphasis on finding shared dimensions between spontaneous and evoked activity subspaces.

2. The effect in figure 3E, F is weak and the data do not fully support the conclusion. Specifically, there seems to be no significant difference between 4dpf and 15dpf in 3E. It is unclear, whether correction for multiple comparisons was used. An alternative presentation would be a matrix wise representation of p-values between all pairs of days.

3. The reported number of components required to explain 80% of the variance for EA (Figure 3C) is high (and increases with age?) compared to the number of stimuli and assemblies found (2C). How can it be reconciled that the number of assemblies is roughly 6, while the number of PC required to explain 80% of the variance is about 40? Also, there is a possible concern here that this number shows a positive bias due to the inclusion of components representing noise. If this high number of components is used to describe the hyperplanes in Figure 3E-G, a concern for bias arises.

4. In Figure 1F, despite a visible trend, the authors report no significant change in the Hamming distance of binarized correlation matrices for SA and TEA. However, it is unclear whether a Hamming distance is the most appropriate measure here (as opposed to e.g. second-order correlation between correlation matrices).

5. Visual system develops earlier than other sensory systems in zebrafish. Therefore, it is likely that major rearrangements to be observed during early development are already established at 4dpf animals. Are there evidence that such features of spontaneous and evoked activity are different in younger animals (2,5dpf) that are just hatched? It is likely that this 2,5dpf represents a stage closer to earlier developmental stage of mammalian visual system.

6. SA and TEA seem to have very different spatial distribution (SA covering larger spaces). If this is true this is rather interesting feature, as evoked activity might change such spatial features of ensembles. Can this be something interesting to analyze, is this a general feature that is stable across development?

7. Is it possible that the evoked patterns appear to get more dissimilar to spontaneous patterns on average, because the spontaneous patterns become noisier over development, as new neurons are added to the network? Is there a way to evaluate the impact of changing noise levels (both recording noise, but also neural noise) in these comparisons?

8. Also is it possible that simply by imaging only a short period of time, spontaneous activity patterns do not really capture all possible combinations of ensembles, whereas as of now the evoked activity recordings are substantially longer then spontaneous activity recordings. How much the recording duration influence such comparisons ensembles during evoked and spontaneous period. To what extent the numbers of captured ensembles depends on the recording duration ?

9. I am surprised that the authors did not discuss some of their results in the context of the findings from a recent paper on the spontaneous and evoked activity in developing zebrafish habenula (doi: https://doi.org/10.1126/sciadv.aaz3173 ), which is another brain region that exhibit both spontaneous and sensory driven activity (DOI: 10.1016/j.cub.2014.01.015 ). In this study the authors showed that at the animals develop spontaneous activity changes with higher correlations between neurons and altered temporal features. Interestingly this study also shows that the spontaneous activity is a good predictor of sensory responses of neurons, and this gets better over development, which in line with the hypothesis that the spontaneous activity might indeed be a prior to evoked responses, at least for some brain regions. I think that authors should at least consider comparing their results and discuss this relationship with these earlier works in zebrafish habenula, in addition to their description of ferret visual cortex work.

---

## [Author Response]

Essential revisions:1. The part explaining the method used in figure 3 E-G should be expanded to ensure the readers can replicate the method. Specifically, it is unclear how many dimensions are necessary to describe the hyperplane H. How many basis vectors were used for the projection? Using many dimensions for the projections could bias the results towards larger angles between patterns and the hyperplane. The authors refer to [23] as the basis of their method. However, the method used in [23] is different from the method described here and puts an emphasis on finding shared dimensions between spontaneous and evoked activity subspaces.

We apologise for implying that our method is based on ref 23; actually it is different. We have now expanded this part in the Methods section, provided examples and added a schematic panel to Figure 3 detailing the technique. The projection on *H* allows us to decompose any pattern into a part which can be "explained" by *H* and a part which can be "explained" by the orthogonal complement of *H*. To find the basis for each space (representing each epoch), we used an equal number of patterns per epoch and set it to be the minimum number of patterns across epochs (mean and std of the minimum number of patterns is 27 and 12 respectively). Thus there is no selection based on variance explained as in [23].

2. The effect in figure 3E, F is weak and the data do not fully support the conclusion. Specifically, there seems to be no significant difference between 4dpf and 15dpf in 3E. It is unclear, whether correction for multiple comparisons was used. An alternative presentation would be a matrix wise representation of p-values between all pairs of days.

We thank the Reviewer for this comment, and we have revised these panels (now 3F,G). We now pool the young fish (4 and 5 dpf) and compare them against the 15 dpf fish group (a single statistical test), which shows a significant developmental effect. The alternative of a matrix-wise representation across all ages would require a large correction for multiple comparisons which will eliminate significance.

3. The reported number of components required to explain 80% of the variance for EA (Figure 3C) is high (and increases with age?) compared to the number of stimuli and assemblies found (2C).

We thank the Reviewer for this point, indeed the number of components required to explain 80% of the variance for EA (Figure 3C) increases with age. We have now added this information in the figure legend.

How can it be reconciled that the number of assemblies is roughly 6, while the number of PC required to explain 80% of the variance is about 40?

These are fundamentally different questions about the data, and so the answers need not be the same. The assembly detection algorithm we used detects groups of neurons that are consistently active together (Avitan et al. 2017, Mölter et al. 2018). It does not include neurons which may contribute high variance overall but are not organised into recurring groups. While there have been PCA-based approaches suggested for detecting assemblies, a rigorous comparison of assembly detection algorithms on synthetic data with known ground truth showed that PCA-based techniques are not as effective for extracting neural assemblies as the Bayesian graph-based method we used (Mölter et al. 2018).

Also, there is a possible concern here that this number shows a positive bias due to the inclusion of components representing noise. If this high number of components is used to describe the hyperplanes in Figure 3E-G, a concern for bias arises.

We do not think there is a clear way of defining ‘noise’ in our data, and thus it is not possible to say which PCs purely represent ‘noise’. In any case, the hyperplanes in Figure 3E and 3F (now 3F and 3G) are not based on the PC space, but on the orthogonal basis spanning the space. We have now detailed the technique in the Methods section, added an example, and provided a schematic panel to Figure 3 to explain the method.

4. In Figure 1F, despite a visible trend, the authors report no significant change in the Hamming distance of binarized correlation matrices for SA and TEA. However, it is unclear whether a Hamming distance is the most appropriate measure here (as opposed to e.g. second-order correlation between correlation matrices).

We thank the Reviewer for the comment, and agree with the suggestion to use the second-order correlation between correlation matrices instead. We have replaced Figure 1F with this, which we agree makes the point more clearly. Main text was updated accordingly (lines 77-81).

5. Visual system develops earlier than other sensory systems in zebrafish. Therefore, it is likely that major rearrangements to be observed during early development are already established at 4dpf animals. Are there evidence that such features of spontaneous and evoked activity are different in younger animals (2,5dpf) that are just hatched? It is likely that this 2,5dpf represents a stage closer to earlier developmental stage of mammalian visual system.

Retinal axons leave the retina at 48 hpf, and the first tectal responses are observed at 3 dpf (Niell and Smith, 2005). We imaged the time window between 4 to 15 since it has been demonstrated that tectal spontaneous activity substantially reorganizes between 5 to 9 dpf, and is also affected by the animal’s visual experience (Avitan et al. 2017). Additionally, between 4-15 dpf evoked tectal activity shows spatial and functional changes, which are closely linked to the behavior (Avitan et al. 2020). It is difficult to make direct comparisons with the timing of mammalian visual development, but we feel that for zebrafish 4-15 dpf clearly constitutes a good developmental window to study the development of the relation between evoked and spontaneous activity.

6. SA and TEA seem to have very different spatial distribution (SA covering larger spaces). If this is true this is rather interesting feature, as evoked activity might change such spatial features of ensembles. Can this be something interesting to analyze, is this a general feature that is stable across development?

We thank the Reviewer for this interesting suggestion, and we have now measured the area covered by the smallest polygon bounding SA and EA assemblies over development. While there is no developmental trend, the area covered by SA assemblies is indeed greater than EA assemblies. These results have now been added to the supplementary materials (Figure S2A and S2B) and mentioned in the main text (Lines 89 and 90).

7. Is it possible that the evoked patterns appear to get more dissimilar to spontaneous patterns on average, because the spontaneous patterns become noisier over development, as new neurons are added to the network? Is there a way to evaluate the impact of changing noise levels (both recording noise, but also neural noise) in these comparisons?

We thank the Reviewer for this comment. Previously we examined the nature of tectal spontaneous activity over development (Avitan et al. 2017) and showed that the frequency of activity peaks at 5 dpf and stabilizes afterwards (7-9 dpf). We have also examined evoked response variability over development (Avitan et al. 2020) and showed that tuned neurons do not change the width of their tuning curve over development. Newly added neurons do indeed add noise to the system (BoulangerWeill et al. 2017) as they are weakly tuned. However, in our analysis we included only tuned neurons. We have now added more information about how neurons were selected to the Methods section (lines 248-249).

8. Also is it possible that simply by imaging only a short period of time, spontaneous activity patterns do not really capture all possible combinations of ensembles, whereas as of now the evoked activity recordings are substantially longer then spontaneous activity recordings. How much the recording duration influence such comparisons ensembles during evoked and spontaneous period. To what extent the numbers of captured ensembles depends on the recording duration ?

We thank the Reviewer for pointing out this potential problem. To address this point we have now repeated the analysis including only the first 30 min of the TEA part of the recording, matching the length of time of the SA part of the recording. These results confirm that the dissimilarity effect over development does not depend on the length of the recording, and are now reported in Figure S3. (Indeed, the p values indicate higher significance than for the results in the main text.)

9. I am surprised that the authors did not discuss some of their results in the context of the findings from a recent paper on the spontaneous and evoked activity in developing zebrafish habenula (doi: https://doi.org/10.1126/sciadv.aaz3173 ), which is another brain region that exhibit both spontaneous and sensory driven activity (DOI: 10.1016/j.cub.2014.01.015 ). In this study the authors showed that at the animals develop spontaneous activity changes with higher correlations between neurons and altered temporal features. Interestingly this study also shows that the spontaneous activity is a good predictor of sensory responses of neurons, and this gets better over development, which in line with the hypothesis that the spontaneous activity might indeed be a prior to evoked responses, at least for some brain regions. I think that authors should at least consider comparing their results and discuss this relationship with these earlier works in zebrafish habenula, in addition to their description of ferret visual cortex work.

We thank the Reviewer for pointing out this interesting paper, which appeared online in its final form after our manuscript was submitted. Although not directly testing the developmental effect, the authors show that the similarity of neurons based on evoked activity to clusters defined by spontaneous activity is significantly different from a shuffle control at 21 dpf but not 3 dpf. We have now included a citation to this paper in the Discussion.